# Aircrew Health: A Systematic Review of Physical Agents as Occupational Risk Factors

**DOI:** 10.3390/ijerph20105849

**Published:** 2023-05-17

**Authors:** Ana Carolina Russo, Elaine Cristina Marqueze, Mariana Souza Gomes Furst, Erika Alvim de Sá e Benevides, Rodrigo Caoduro Roscani, Celso Amorim Salim, Paulo Cesar Vaz Guimarães

**Affiliations:** Fundação Jorge Duprat Figueiredo de Segurança e Medicina do Trabalho—Fundacentro, São Paulo 30180-100, Brazil

**Keywords:** systematic review 1, aircrew2, occupational risk factors

## Abstract

The primary objective of this systematic review was to analyze the main physical agents representing risk factors for commercial aircrew, together with their consequences. The secondary objective was to identify the countries in which studies on the topic were conducted, as well as the quality of the publications available. Thirty-five articles, published between 1996 and 2020, were selected for the review, having met all inclusion criteria. The majority of studies were conducted in the United States, Germany, and Finland and had moderate or low methodological quality of evidence. The main risk factors for aircrew identified in publications were exposure to abnormal air pressure, cosmic radiation, noise, and vibrations. Hypobaric pressure was explored in response to demands for studies on this agent, a factor which may lead to otic and ear barotraumas, as well as acceleration of atherosclerosis of the carotid artery. However, there is a dearth of research exploring this phenomenon.

## 1. Introduction

The occupational activity of aircrew has specific characteristics inherent to the job. The physical risk agents that potentially affect these workers include air pressure, noise, vibration, ionizing radiation, and temperature [1]. The aviation environment presents numerous challenges to humans, such as the lack of time to adapt to rapid hypobaric changes. In this context, aviation medicine should be preventive in nature and recognized as a public health issue.

Sporadic or prolonged exposure to these agents can damage the health of workers. One of the most common health problems associated with the aviation profession is middle-ear barotrauma, defined as inflammation of the tympanic membrane and middle-ear cavity due to differences in pressure between the cavity and external atmospheric pressure, a phenomenon also associated with diving or hyperbaric oxygen therapy [2,3].

Changes in atmospheric pressure in the cabin can lead to barotrauma, uni- or bi-lateral, in the first few years on the job [4]. Because of this risk, crew members are advised not to fly when exhibiting upper respiratory tract symptoms [5]. It is important to highlight that in 95.4% of barotrauma cases, the events occurred at the time of descent of the aircraft from cruising to landing altitudes [6]. 

In addition to the possibility of barotrauma [7], hypobarism can decrease and compromise mental capacity, including judgment, memory, and performance of fine motor movements [8].

Another pressure-related problem is hypoxia, which occurs during cabin decompression. The low oxygen concentrations can lead to drowsiness, headache, dizziness, visual changes, cyanosis, hilarity, euphoria, convulsions, spasms [9], and altered cerebral white matter [10,11].

In addition to the implications of pressure, there are potential problems associated with ionizing radiation [12], which pose an additional risk to the health of aircraft crew members [13]. The effects of this agent include gene mutations, breakdown of the DNA molecule, and generation of free radicals, which increase the risk of cancer in exposed individuals and their offspring, while also promoting cardiovascular disorders, cataracts, and genetic alterations [14].

This topic has attracted attention from Brazilian trade unions, universities, and researchers, representing a new field of research with multiple facets and possibilities. Although the subject has been addressed through interdisciplinary approaches from different areas of science, such as engineering and health, studies exploring the occupational impact of physical agents (especially ionizing radiation and hypobaric pressure) on Brazilian aircrew are still scarce and therefore deserve attention. 

This dearth of investigations prompted the present study, the objective of which was to analyze and elucidate the main occupational risks to pilots and flight attendants from exposure to physical agents in the work environment. This is the first study of its kind identifying, categorizing, and analyzing the current evidence reported in the available literature on the risk of physical agents among this category of workers. This study also serves to highlight the lack of Brazilian research in this population.

## 2. Materials and Methods

This theoretical study involved the application of the technical procedure for a systematic literature review (SLR). This technique was used to identify, evaluate, and interpret the relevant research on occupational physical agents present in civil aviation using a defined methodological sequence that allows knowledge to be pooled and built [15,16].

The design of this SLR was devised according to the guidelines of the Preferred Reporting Items for Systematic Reviews and Meta-analyses (PRISMA) statement [17] and was registered on the PROSPERO platform (number 240012). Given that the study was a literature review, submission to the Research Ethics Committee for prior approval was waived.

The SLR comprises a sequence of three stages: planning, conducting, and presenting the review, each of which entails its respective actions (Figure 1).

### 2.1. Bibliometric Mapping

The VOSviewer software was employed (using terms and keywords) to map the main activities in the area of study (Occupational Risks × Aircrew) and visualize bibliometric maps from the review.

The resulting data were analyzed using VOSviewer software to build relationship networks of terms from the Web of Science, Scopus, PubMed, and Cochrane databases used in the systematic literature review.

### 2.2. Search Strategy

The PICo strategy for non-clinical research was used to construct the research question (Table 1), namely: Are the physical agents of ionizing and cosmic radiation, abnormal air pressure, and noise occupational risk factors for commercial aircrew?

The planning phase began based on a review of possible occupational risks involving aircrew to gain greater familiarity with the theme addressed and better define the research problem. In this phase, the feasibility and necessity of this study were also confirmed, mainly by drawing on previous reports [4,11,12], which served as the main theoretical basis of the investigation. 

The searches were carried out on the Web of Science, Scopus, Pub Med, and Cochrane databases from February to May 2021, chosen for their interdisciplinary nature and for constituting the largest databases available. The keywords were defined according to the Medical Subject Headings (MeSH).

Initially, a search using the key words “Risk Factors” and “Aircrew” was conducted on the Web of Science to identify the publication period to be considered (Figure 2).

The first publications on the subject began around 1992 and, therefore, the period from 1990 to 2021 was adopted for searches (Figure 2).

The main areas of knowledge related to the theme, obtained using the search terms “Risk Factors” and “Aircrew”, are described in Figure 3. All relevant publications were in English; therefore, this language was established as a filter for the search.

The occupational health area had the highest number of studies on the occupational risks of aircrew.

Based on the definitions of the terms to be searched, the following combinations of key words were obtained, resulting in 24 permutations (Figure 4):

### 2.3. Eligibility Criteria

Potentially relevant studies were selected by two independent reviewers according to the following inclusion criteria: (1) articles in Portuguese and English; (2) articles available in full; (3) only studies with observational design (cross-sectional, cohort, case-control) and interventions; (4) articles published since 1990; (5) studies involving men and women of any age group; (6) studies evaluating the main occupational physical risks; (7) peer-reviewed articles; and (8) only investigation of civil aviation. 

Studies whose design type was clinical trials, ecological studies, review studies or qualitative studies were excluded, as were studies involving retired professionals, studies with different objectives from those of the present review, studies involving helicopter pilots, abstracts, technical reports, oral communications, letters to the Editor, studies with simulated flights, and studies of military aircrew and helicopter studies.

The initial selection of articles was performed independently by the reading of titles and abstracts. Subsequently, both reviewers read the full texts of the articles that met the inclusion criteria. Any disagreements regarding the eligibility of the articles were settled by consulting a third researcher. The number of studies included and excluded in the different phases of the systematic review is presented in the PRISMA flowchart (Table 2).

### 2.4. Article Data Extraction

Data extraction included the following variables: author, year, study design, sample, objective, and main study results. Microsoft Office Excel 2021^®^ (version 2302) was used to tabulate the parameters extracted.

### 2.5. Quality Assessment

To evaluate the risk of bias and methodological quality of the studies reviewed, two tools were used: the Newcastle–Ottawa Scale [18] for cohort and case-control studies, and the guidelines of Loney et al. (1998) for cross-sectional studies. 

As described in the Newcastle–Ottawa Scale, the methodological quality score of cohort and case-control studies was calculated based on three components (ranging from 0 to 9 points): (1) group selection (0–4 points); (2) quality of adjustment for confounders (0–2 points); and (3) exposure evaluation after outcome (0–3 points) [18]. Cross-sectional studies, in turn, were evaluated using Loney’s criterion, yielding a range of 0 to 8 points, with the highest scores indicating higher methodological quality. The score is obtained from the eight questions that make up the scale, and each of the questions evaluated is assigned one or zero according to the adequacy of the methods and presentation of the results [19].

## 3. Results

The search strategy led to the identification of 3938 articles. A total of 1542 duplicate articles were removed. Of the remaining studies, 1288 were selected for title and abstract screening, of which 1181 were excluded for not meeting the inclusion criteria. 

Of the 107 remaining articles evaluated, 72 were excluded for not meeting the inclusion criteria. Therefore, 35 articles were included in this systematic review. A consensus between the two reviewers was reached for all of the articles included (Figure 5).

### 3.1. Bibliometric Analysis

Graphs were generated based on the search terms (Figure 4). The two analyses were combined by constructing two distinct maps (co-authorship and keywords). Thus, the keywords used with greater frequency and intensity, as well as the co-authorship relationships were analyzed.

Using the VOSviewer software (version 1.6.16), the main words in the titles and abstracts of the articles in the selected databases (Web of Science, SCOPUS, PubMed, and Cochrane) were detected. Words were detected based on a co-occurrence network, as shown in Figure 6. Seven clusters were formed using the VOSviewer grouping technique. The words presented in each cluster were related to different areas of research.

The colors shown in Figure 6 are random, serving to distinguish the s-work groups. Each circle represents a term, with names truncated by the software to avoid visual overlaps [20]. Circle size reflects the number of occurrences of a term. The words present in the clusters have a direct relationship with each other, corresponding to their separation factors. The size of each word in the cluster is related to its weight, i.e., co-occurrence in publications.

The map highlights the terms “mortality”, “cancer”, “cosmic radiation”, “aviation”, “altitude”, “risk”, and “pilots” as being the main concepts of established relationships. 

The last analysis in VOSviewer sought to identify the networks of co-authorship among authors related to the topic, i.e., the inter-relationship of the authors (Figure 7).

The analysis indicated the existence of 125 clusters or groups of authors with affinity in their research. However, 12 of the 125 groups had a greater influence (Figure 8).

### 3.2. Where Were Studies Conducted? What Is Brazil’s Output?

Figure 9 shows the origins of the articles reviewed. Most of the investigations were carried out in the United States (29%), followed by Germany (17%) and Finland (17%). No public Brazilian research was identified, evidencing the need for more local studies.

Among the studies involving flight attendants published between 1999 and 2018, 15% were conducted in the last five years (2017–2021) (Table 3). In studies of pilots published between 1996 and 2018, only 7% date back to the last five years (Table 4). Finally, the studies involving both flight attendants and pilots were published between 2001 and 2020, 28% of which were carried out in the last five years (Table 5). In short, of the total studies reviewed, 29% were conducted between 2017 and 2020.

#### 3.2.1. Studies Involving Flight Attendants

The analysis of the main results found in studies of flight attendants showed that exposure to cosmic radiation was the leading risk factor. Given this risk factor, cancer was the main outcome (Table 3).

Three studies [21,26,29] investigated whether exposure to cosmic radiation caused biological effects in flight attendants. The case-control study failed to identify biological alterations relative to the rest of the population (Table 3).

The study by Haldorsen et al. [22] evaluated the incidence of cancer as a function of cosmic radiation in 3693 cabin attendants. Among women, a higher incidence of breast cancer was observed, while among men, a higher risk of cancer was observed in the upper respiratory and gastric tract. Additionally, the study by Pukkala et al. [27] of 8507 female and 1559 male airline cabin attendants found significant excesses in skin melanoma among men. The study by Mcneely [32] observed an association between job tenure and breast cancer among women who had three children or more children (Table 3).

Rafnsson et al. [24] studied 1532 flight attendants and found that length of employment was associated with risk of breast cancer. However, the study by Paridou et al. [23] failed to identify a pattern of cancer with duration of employment in an analysis of 1835 flight attendants. Three studies [25,31,33] found no statistically significant association between cumulative radiation dose and cancer, whereas family history of cancer had a stronger influence on occurrence of the disease (Table 3).

Lastly, a cohort study [28] of 2273 flight attendants investigating whether exposure to cosmic radiation was associated with cases of miscarriage concluded that the risk of cases was not increased compared with the general population (Table 3).

#### 3.2.2. Studies Involving Pilots

The most investigated themes among pilots were cancer due to exposure to cosmic radiation, and barotrauma due to exposure to abnormal air pressure (Table 4).

The study of Band et al. [34] evaluated the incidence of cancer as a function of occupational radiation exposure in 2740 pilots. The authors concluded that long-term follow-up is required for a more in-depth evaluation of cancer incidence in this population. In the study by Haldorsen et al. [36] investigating 3701 pilots, elevated risks were found for malignant melanoma and non-melanoma skin cancer and, according to 2 articles [37,41,43], this risk can be increased due to the greater number of hours of long-haul flights. In addition to flying hours, the study by Langner et al. [39] found that childhood sunburn and a family history of skin cancer were conditions that increased the risk of cancer (Table 4).

However, based on the results of the studies by [35,46], no cancer mortality risk was found to be substantially increased due to exposure to ionizing radiation compared to the rest of the population (Table 4).

Two studies [40,44] evaluated whether exposure to cosmic radiation contributes to an increase in chromosomal aberrations. The results confirmed this hypothesis; however, the authors highlighted the need for studies involving a larger number of participants. Complementing this research, Rosenkvist et al. [42] studied 2802 pilots to determine the frequency of translocations and insertions in the blood over the long-term with respect to radiation dose received. The results showed that the mean number of translocations per cell was significantly higher among pilots than in controls (Table 4).

Papailiou [45] investigated 4018 pilots and identified that cosmic radiation, in addition to the above-mentioned effects, can influence the diastolic and systolic blood pressures of these professionals (Table 4), thus altering cardiovascular functionality.

In addition to cosmic radiation, air pressure is another relevant factor. Two studies [38,48] evaluated the effects of exposure to abnormal pressures on pilots. The first [38] identified several cases of barotrauma, mainly affecting the ear. The authors of the second study [48] observed acceleration of carotid artery atherosclerosis in pilots (Table 4).

Based on the results of studies in pilots, the investigations analyzing exposure to cosmic radiation, although more numerous, are inconclusive on whether cancer risk is higher in pilots than in the rest of the population. By contrast, articles on exposure to abnormal pressures, albeit numbering only two, present evidence supporting damage caused to pilot health (Table 4).

#### 3.2.3. Studies Involving Both Attendants and Pilots

In studies conducted with both flight attendants and pilots simultaneously, it was found that, similarly to the results of studies conducted solely with pilots, the risk factors associated with cosmic radiation and abnormal pressures stand out (Table 5).

One study [49] identified an increased risk of cancer, especially for those who had a higher cosmic exposure at a young age and those hypersensitive to bleomycin [51]. However, in several studies [50,52,54], the authors failed to find a higher risk of cancer in this population, and it was actually lower in some cases because the subjects did not present smoking-related cancer.

Regarding exposure to abnormal air pressure, one study [52] explored the prevalence of otic barotrauma and its risk factors in crew members. The authors found that the risk of barotrauma in these professionals increased for individuals who were smokers, presented allergy, hormonal disorder, and/or worked over 70 h per month.

It is important to highlight that none of the studies included in this review evaluated ways of mitigating the risk factors investigated. The authors only presented suggestions for what can be conducted in an effort to minimize health risks, highlighting reducing consecutive working days and allowing longer rest time between work shifts, especially on long-haul flights.

### 3.3. Quality Assessment of Studies Reviewed

Of the 35 articles reviewed, 12 were conducted for flight attendants (37.1%) only, 15 for pilots (42.9%), and 7 evaluated both flight attendants and pilots concomitantly (20.0%) (Table 6). Of the 12 studies of flight attendants alone, 11 were cohort studies (91.7%) and 1 was a case-control study (8.3%). Only one of the cohort studies attained a score of 6 points, i.e., had strong evidence; nine scored 4 points, and two scored 3 points, indicating that most of the studies reviewed had moderate evidence (Table 6).

Of the studies conducted among pilots (n = 16) only, the majority were retrospective (n = 14, 87.5%). Of the 14 retrospective studies, 9 scored 4 or 5 points, indicating moderate evidence; 7 scored 4 points and 2 attained 3 points, suggesting moderate quality. Five scored only 3 points, indicating limited evidence. The cohort study (6.25%) and case-control study (6.25%) both scored 3 points, indicating limited evidence (Table 7).

In the case of the 7 studies involving both flight attendants and pilots concomitantly, the majority were retrospective cohort studies (85.7%). Of these cohort studies, four had moderate evidence, two scored 4 points, two 5 points, and two studies attained 3 points, indicating limited evidence. Only the cross-sectional study (14.3%) obtained a score of 5 points, indicating moderate quality (Table 8).

## 4. Discussion

The most prevalent physical health risk factors among flight attendants and pilots were ionizing radiation and abnormal air pressure. Secondary to these risk factors, the most prevalent health problems associated with ionizing radiation in the articles analyzed were problems related to cancer [22,24,27,32,36,39], risks in the reproductive system [28], and cardiovascular functionalities [47].

However, most authors stated the p effects in aircrew were similar to those in the non-exposed general population [23,25,30,31,35,37,43,54]. In order to gain a better understanding of the risks, long-term monitoring of crew members [28] and assessment of a larger number of participants [44] are required.

For exposure to abnormal air pressure, the most prevalent outcomes were barotrauma [38,53] and changes in blood pressure and cerebral blood flow [48]. This change in blood pressure can be caused by hypoxia, characterized by low oxygen concentrations in cellular tissues. In aviation, this symptom is caused when the altitude increases, accompanied by decrease in atmospheric pressure and consequent decrease in partial oxygen pressure, known in this case as hypoxic hypoxia [56].

Although the percentage of oxygen in the atmosphere is constant, the decrease in partial pressure reduces the differential pressure between the blood flow in the pulmonary alveolus and atmospheric oxygen, thus hindering gas exchange between them.

Regarding the quality of the studies analyzed in the present review, most were cohort studies with a moderate level of evidence (63%), i.e., there was moderate confidence in the estimated effect. However, these studies have made important contributions to the understanding of the main risk factors for aircrew health.

The absence of Brazilian studies in the articles retrieved is also noteworthy. This highlights the need for further studies to address the issue of occupational health among Brazilian aircrew and for further discussion on the development of public policies in this sector, especially given that Brazil has one of the largest aircraft fleets in the world. 

However, the current reality is that aircrew health protection policies remain largely ineffective [56], failing to consider, for example, exposure to agents such as hypobaric pressure. Therefore, exposure to physical risk agents may promote the onset and/or worsening of diseases [57] which, over time, can become chronic, as documented in previous Brazilian studies [12], some of which are centered on aerospace medicine [5].

The study by Loterio et al. [7] underscores the need to restructure the working conditions offered to Brazilian aircrew.

## 5. Conclusions

This systematic review explored the available literature on occupational risks arising from physical agents in both pilots and flight attendants. Many studies have reported effects on cancer incidence of exposure to ionizing radiation. However, the results obtained in the studies reviewed showed no significant difference between cancer cases in aircrew and the general population.

Although in the minority, studies on the risks of exposure to abnormal air pressures, especially hypobarics, have yielded relevant findings. Hypobaric pressure may lead to ear and otic barotrauma and acceleration of carotid artery atherosclerosis. However, there is a dearth of research examining the phenomenon.

Regarding quality, most studies on risk factors for aircrew were rated as being of moderate or limited methodological and evidence quality. The main limitations pointed out in the studies were the short analysis period and the low number of participants in the research. 

No Brazilian articles were found among the studies reviewed, reinforcing the need for national studies investigating the specificities of work in Brazil since characteristics inherent to the country such as climate, labor laws, etc., can influence working conditions, differentiating them from aircrew in other locations. 

There is a need for further research to further and deepen scientific knowledge on the subject. This research should directly involve professionals in the area and adopt a quantitative approach to yield relevant statistical data. Future studies could entail an experimental approach and field research; for example, by including medical examinations that monitor health problems and confirm their causality.

Thus, further studies should be conducted on occupational risks, preferably with longitudinal designs, so as to provide more robust evidence of the risk factors linked to physical agents and evaluate ways of reducing them, thereby enhancing quality of life in the workplace for this group of Brazilian professionals.

## Figures and Tables

**Figure 1 ijerph-20-05849-f001:**
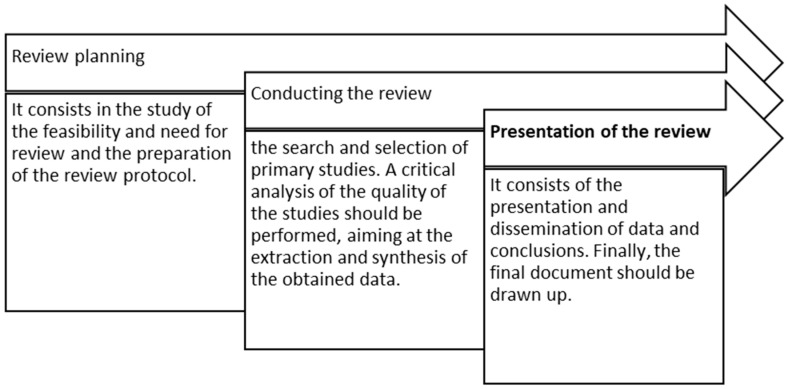
Stages of the systematic literature review. (Adapted with permission from Ref. [15]).

**Figure 2 ijerph-20-05849-f002:**
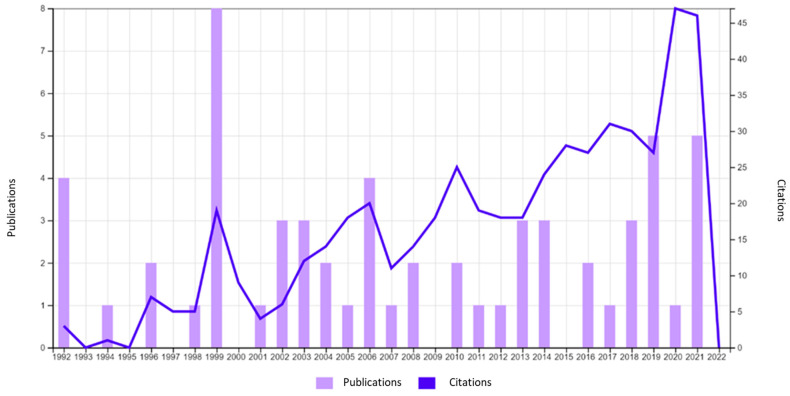
Publications cited by year (Web of Science).

**Figure 3 ijerph-20-05849-f003:**
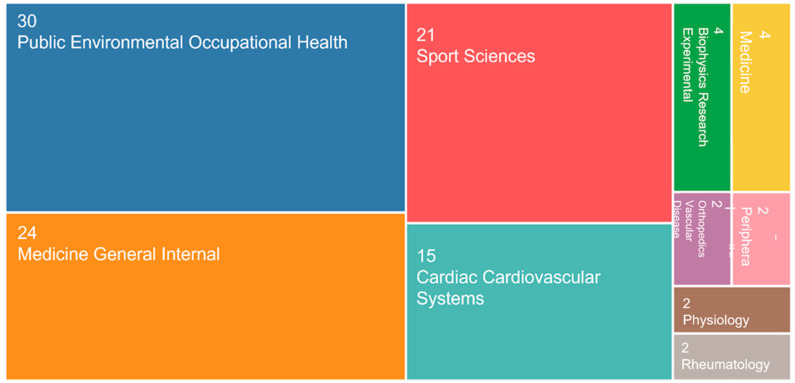
Main areas of knowledge (Web of Science).

**Figure 4 ijerph-20-05849-f004:**
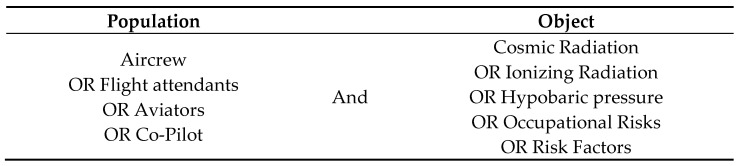
Combination of key words used in the search.

**Figure 5 ijerph-20-05849-f005:**
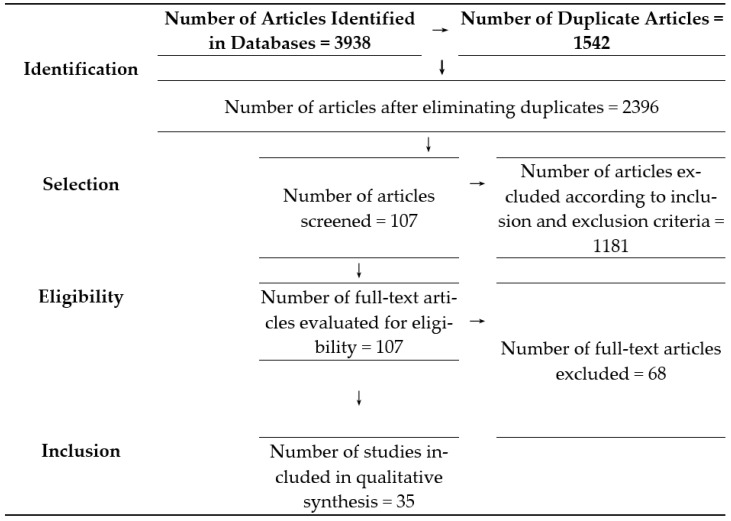
Flow diagram of study selection process for inclusion in systematic review.

**Figure 6 ijerph-20-05849-f006:**
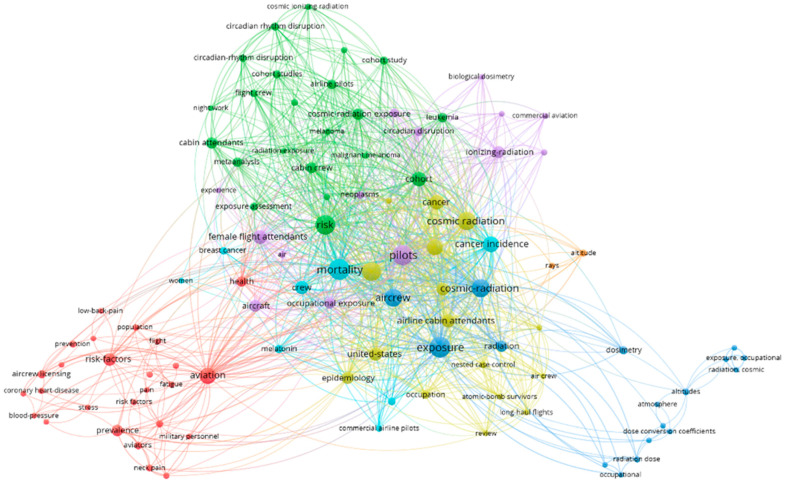
Keywords used by authors (7 clusters formed).

**Figure 7 ijerph-20-05849-f007:**
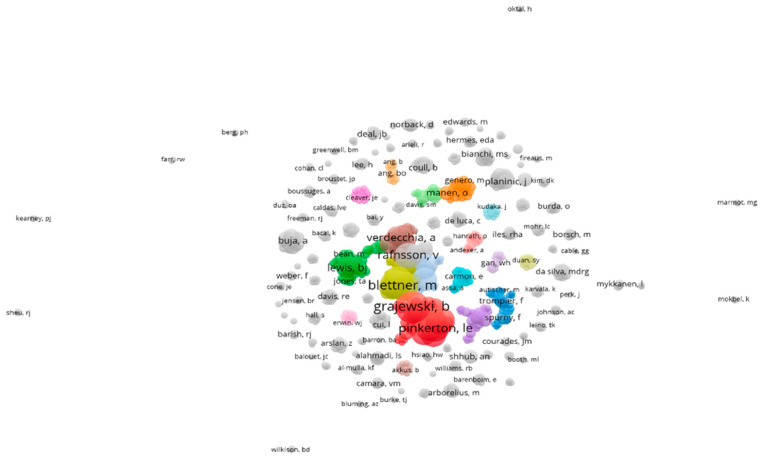
Clusters identified in searches.

**Figure 8 ijerph-20-05849-f008:**
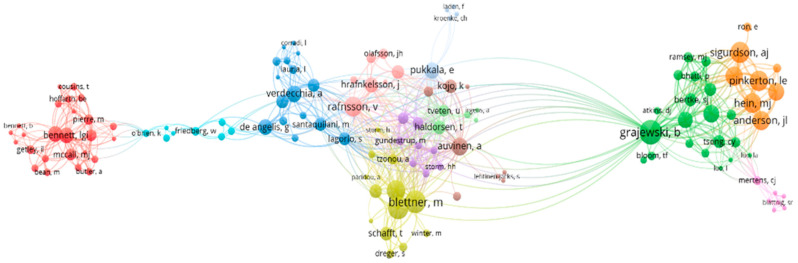
Formation of 12 clusters.

**Figure 9 ijerph-20-05849-f009:**
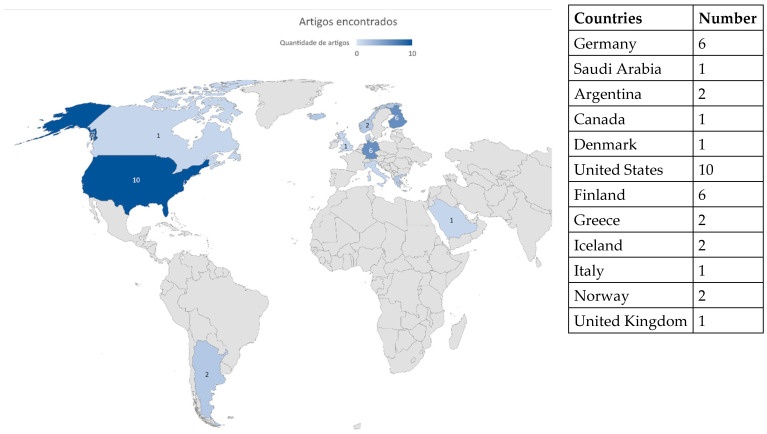
Origin of articles reviewed.

**Table 1 ijerph-20-05849-t001:** PICo strategy for the elaboration of the research question.

Criterion	Definition
Population	Aircrew (Pilots and Flight Attendants)
Interest	Exposure to physical agents (ionizing radiation, cosmic radiation, abnormal air pressures, and vibration)
Context	Occupational risk factor

**Table 2 ijerph-20-05849-t002:** PRISMA checklist.

Section and Topic	Item #	Checklist Item	Location Where Item Is Reported
**TITLE**			
Title	1	Identify the report as a systematic review.	1
**ABSTRACT**			
Abstract	2	See the PRISMA 2020 for Abstracts checklist.	1
**INTRODUCTION**			
Rationale	3	Describe the rationale for the review in the context of existing knowledge.	1–2
Objectives	4	Provide an explicit statement of the objective(s) or question(s) the review addresses.	2
**METHODS**			
Eligibility criteria	5	Specify the inclusion and exclusion criteria for the review and how studies were grouped for the syntheses.	5
Information sources	6	Specify all databases, registers, websites, organisations, reference lists, and other sources searched or consulted to identify studies. Specify the date when each source was last searched or consulted.	3
Search strategy	7	Present the full search strategies for all databases, registers, and websites, including any filters and limits used.	3–6
Selection process	8	Specify the methods used to decide whether a study met the inclusion criteria of the review, including how many reviewers screened each record and each report retrieved, whether they worked independently, and if applicable, details of automation tools used in the process.	3–6
Data collection process	9	Specify the methods used to collect data from reports, including how many reviewers collected data from each report, whether they worked independently, any processes for obtaining or confirming data from study investigators, and, if applicable, details of automation tools used in the process.	3–6
Data items	10a	List and define all outcomes for which data were sought. Specify whether all results that were compatible with each outcome domain in each study were sought (e.g., for all measures, time points, analyses), and, if not, the methods used to decide which results to collect.	3–6
	10b	List and define all other variables for which data were sought (e.g., participant and intervention characteristics, funding sources). Describe any assumptions made about any missing or unclear information.	3–6
Study risk of bias assessment	11	Specify the methods used to assess risk of bias in the included studies, including details of the tool(s) used, how many reviewers assessed each study and whether they worked independently, and, if applicable, details of automation tools used in the process.	3–6
Effect measures	12	Specify for each outcome the effect measure(s) (e.g., risk ratio, mean difference) used in the synthesis or presentation of results.	3–6
Synthesis methods	13a	Describe the processes used to decide which studies were eligible for each synthesis (e.g., tabulating the study intervention characteristics and comparing against the planned groups for each synthesis (item #5)).	3–6
	13b	Describe any methods required to prepare the data for presentation or synthesis, such as handling of missing summary statistics or data conversions.	3–6
	13c	Describe any methods used to tabulate or visually display results of individual studies and syntheses.	3–6
	13d	Describe any methods used to synthesize results and provide a rationale for the choice(s). If meta-analysis was performed, describe the model(s), the method(s) to identify the presence and extent of statistical heterogeneity, and software package(s) used.	3–6
	13e	Describe any methods used to explore possible causes of heterogeneity among study results (e.g., subgroup analysis, meta-regression).	3–6
	13f	Describe any sensitivity analyses conducted to assess robustness of the synthesized results.	3–6
Reporting bias assessment	14	Describe any methods used to assess risk of bias due to missing results in a synthesis (arising from reporting biases).	3–6
Certainty assessment	15	Describe any methods used to assess certainty (or confidence) in the body of evidence for an outcome.	3–6
**RESULTS**			
Study selection	16a	Describe the results of the search and selection process, from the number of records identified in the search to the number of studies included in the review, ideally using a flow diagram.	6
	16b	Cite studies that might appear to meet the inclusion criteria but which were excluded and explain why they were excluded.	6
Study characteristics	17	Cite each included study and present its characteristics.	9–27
Risk of bias in studies	18	Present assessments of risk of bias for each included study.	9–27
Results of individual studies	19	For all outcomes, present for each study: (a) summary statistics for each group (where appropriate) and (b) an effect estimate and its precision (e.g., confidence/credible interval), ideally using structured tables or plots.	9–27
Results of syntheses	20a	For each synthesis, briefly summarise the characteristics and risk of bias among contributing studies.	9–27
	20b	Present results of all statistical syntheses conducted. If meta-analysis was conducted, present for each the summary estimate and its precision (e.g., confidence/credible interval) and measures of statistical heterogeneity. If comparing groups, describe the direction of the effect.	9–27
	20c	Present results of all investigations of possible causes of heterogeneity among study results.	9–27
	20d	Present results of all sensitivity analyses conducted to assess the robustness of the synthesized results.	9–27
Reporting biases	21	Present assessments of risk of bias due to missing results (arising from reporting biases) for each synthesis assessed.	9–27
Certainty of evidence	22	Present assessments of certainty (or confidence) in the body of evidence for each outcome assessed.	9–27
**DISCUSSION**			
Discussion	23a	Provide a general interpretation of the results in the context of other evidence.	27–28
	23b	Discuss any limitations of the evidence included in the review.	27–28
	23c	Discuss any limitations of the review processes used.	27–28
	23d	Discuss implications of the results for practice, policy, and future research.	27–28
**OTHER INFORMATION**			
Registration and protocol	24a	Provide registration information for the review, including register name and registration number, or state that the review was not registered.	2
	24b	Indicate where the review protocol can be accessed, or state that a protocol was not prepared.	2
	24c	Describe and explain any amendments to information provided at registration or in the protocol.	2
Support	25	Describe sources of financial or non-financial support for the review and the role of the funders or sponsors in the review.	-
Competing interests	26	Declare any competing interests of review authors.	-
Availability of data, code, and other materials	27	Report which of the following are publicly available and where they can be found: template data collection forms; data extracted from included studies; data used for all analyses; analytic code; any other materials used in the review.	-

**Table 3 ijerph-20-05849-t003:** Risk factors and outcomes in studies reviewed involving flight attendants.

Authors (Year)	Objective	Risk Factors	Outcome	Sample
Wolf G, Obe G, Bergau L. (1999) [21]	Determine whether there are biological effects of exposure to low-dose radiation in aircrew.	Exposure to cosmic radiation	Analysis of biological effects	Test group: 59 female cabin attendants with at least 10 years of seniority; full-time job; predominantly on long-haul flights. Site control: 31 women from station staff with at least 5 years of seniority. Positive control: 9 blood samples irradiated in vitro (0.1 to 0.5 Gy)
Haldorsen T, Reitan JB, Tveten U (2001) [22]	Evaluate increased incidence of cancer as a function of cosmic radiation.	Exposure to cosmic radiation	Cancer	3693 flight attendants
Paridou A, Velonakis E, Langner I, Zeeb H, Blettner M Tzonou A (2003) [23]	Study pattern of mortality among airline crew.	Exposure to cosmic radiation	Mortality pattern	1835 flight attendants
Rafnsson V, Sulem P, Tulinius H, Hrafnkelsson J (2003) [24]	Investigate whether length of employment as a cabin attendant was related to breast cancer when adjusted for reproductive factors.	Exposure to cosmic radiation	Breast cancer	1532 cabin attendants
Kojo K, Pukkala E, Auvinen A (2005) [25]	Assess the contribution of occupational versus lifestyle and other factors to breast cancer risk among cabin attendants.	Exposure to cosmic radiation and lifestyle	Breast cancer	1041 cabin attendants
Pinkerton LE, Waters MA, Misty J. Hein; Zachary Zivkovich; Mary K. Schubauer-Berigan; Barbara Grajewski (2012) [26]	Assess mortality among 11,311 former U.S. flight attendants. The primary a priori outcomes of interest were breast cancer and melanoma.	Exposure to cosmic radiation	Breast cancer and melanoma	11,311 former flight attendants
Pukkala E, Helminen M, Haldorsen T; Hammar N; Kojo K; Linnersj€o A; Rafnsson V; Tulinius H; Tveten U, Auvinen Anssi (2012) [27]	Assess the influence of work-related factors on cancer incidence of cabin crew members.	Exposure to cosmic radiation	Cancer	8507 women and 1559 male airline cabin attendants
Grajewski B, Whelan EA, Lawson CC, Hein MJ, Waters MA, Anderson JL, MacDonald LA, Mertens CJ, Tseng CY, Cassinelli RT, Luo L (2015) [28]	Evaluate reproductive risks and circadian disruption as a function of radiation exposure.	Exposure to cosmic radiation	Reproductive risks and circadian disruption	2273 flight attendants and 381 teachers
Schubauer-Berigan MK, Anderson JL, Hein MJ, Little MP, Sigurdson AJ; Pinkerton LE (2015) [29]	Evaluate the association of breast cancer incidence (BCI) with cosmic radiation dose and circadian rhythm disruption.	Exposure to cosmic radiation and disruption of circadian rhythm	Breast cancer	6093 former flight attendants
Johnson CY, Grajewski B, Lawson CC, Whelan EA, Bertke SJ, Tseng CY (2016) [30]	This study aimed to (i) compare odds of endometriosis in a cohort of flight attendants against a comparison group of teachers and (ii) investigate occupational risk factors for endometriosis among flight attendants.	Exposure to cosmic radiation	Endometriosis	1945 flight attendants and 236 teachers aged 18–45 years
Pinkerton LE; Hein MJ; Anderson JL; Little MP; Sigurdson AJ; Schubauer-Berigan MK (2016) [31]	The aim of the study was to examine the association of breast cancer incidence with cosmic radiation dose and circadian rhythm disruption.	Exposure to cosmic radiation	Breast cancer and circadian rhythm disruption	6093 U.S. flight attendants
Mcneely E, Mordukhovich I, Staffa S, Tideman S, Gale S, Coull B (2018) [32]	Characterize the prevalence of cancer diagnoses among U.S. cabin crew relative to the general population.	Exposure to cosmic radiation	Cancer	Flight attendants (n = 5366)
Pinkerton LE, Hein MJ, Anderson JL, Christianson A, Dphil MPL, Sigurdson AJ, Schubauer-Berigan MK (2018) [33]	Assess the incidence of cancer and circadian disruption as a function of occupational radiation exposure.	Exposure to cosmic radiation	Cancer and circadian disruption	6000 female flight attendants compared to the US population

**Table 4 ijerph-20-05849-t004:** Risk factors and outcomes of studies reviewed involving pilots.

Authors (Year)	Objective	Risk Factors	Outcome	Sample
Band PR, Le ND, Fang R, Deschamps M, Coldman AJ, Gallagher RP, Moody J (1996) [34]	Evaluate the incidence of cancer as a function of occupational radiation exposure.	Exposure to cosmic radiation	Cancer	2740 pilots
Romano E, Ferrucci L, Nicolai F, Derme V, Stefano GFD (1997) [35]	Assess whether there is an increase of chromosomal aberrations.	Exposure to cosmic radiation	Increase in chromosomal aberrations	120 males and 72 females occupied in commercial aviation as pilots
Haldorsen T, Reitan JB, Tveten U (2000) [36]	Determine whether exposure at work influences the incidence of cancer in commercial pilots.	Exposure to cosmic radiation	Cancer	3701 male pilots
Pukkala E, Aspholm R, Auvinen A, Eliasch H, Gundestrup M, Haldorsen T, Hammar N, Hrafnkelsson J, Kyyrönen P, Linnersjö Anette, Rafnsson V, Storm H, Tveten U (2002) [37]	Assess the incidence of cancer among male airline pilots in the Nordic countries, with special reference to the risk related to cosmic radiation.	Exposure to cosmic radiation	Cancer	10,032 male airline pilots, withaverage follow-up of 17 years
Zeeb; Blettner; Langner (2002) [38]	Assess the influence of occupational and other factors on mortality.	Exposure to cosmic radiation	Cancer	6061 male cockpit personnel
Langner I, Blettner M, Gundestrup M, Storm H, Aspholm R, Auvinen A, Pukkala E, Hammer GP, Zeeb, Hrafnkelsson J, Rafnsson V, Tulinius H, Angelis GD, Verdecchia A, Haldorsen T, Tveten U, Eliasch H†, Hammar N; Linnersj A (2004) [39]	Evaluate cancer incidence as a function of occupational radiation exposure.	Exposure to cosmic radiation	Cancer	19,184 male pilots
Nicholas JS, Butler GC, Davis S, Bryant E, Hoel DG, Mohr Jr. LC (2003) [40]	Determine the frequency of translocations and insertions in the blood of long-term pilots in relation to estimated cumulative radiation dose received while flying, and to compare this to the frequency in a group of aged men without a history of frequent airline travel.	Exposure to cosmic radiation	Translocations and insertions in the blood	2802 in pilots and 3000 in controls
Pukkala E, Aspholm R, Auvinen A, Eliasch H, Gundestrup M, Haldorsen T, Hammar N, Hrafnkelsson J, Kyyro P, Rafnsson V, Storm H, Tveten U (2003) [41]	Assess cancer incidence through national cancer registries.	Exposure to cosmic radiation	Cancer	Pilots: 10,051 male and 160 female airline pilots
Rosenkvist L, Klokker M, and Katholm M (2008) [42]	Analyze the incidence of barotrauma in pilots.	Exposure to abnormal pressures	Barotrauma	948 commercial pilots
Nicholas JS, Swearingen CJ, Kilmer JB (2009) [43]	Evaluate the incidence of skin cancer in pilots as a function of occupational radiation exposure.	Exposure to cosmic radiation	Skin cancer	2428 pilots
Luca JC, Picco SJ, Macintyre C, Dulout FN, Lopez-Larraza DM (2010) [44]	Analyze the effects of chronic exposure of Argentine crew members to low doses of ionizing radiation.	Exposure to cosmic radiation	Effects of chronic exposure	Technical ground workers (group A; n = 10), domestic flight pilots (group B; n = 14), trans-equatorial flight pilots (group C; n = 17), transpolar flight pilots (group D; n = 17) and retired pilots (group E; n = 10)
Papailiou M, Mavromichalaki H, Kudela K, Stetiarova J, Dimitrova S (2012) [45]	Examine the potential effects of cosmic radiation on the cardiovascular functionality of a group of Slovak aviators.	Exposure to cosmic radiation	Cardiovascular functionality	4018 pilots
Kojo K, Helminen M, Pukkala E, Auvinen A (2013) [46]	Evaluate whether the difference in risk factor prevalence between Finnish airline cabin crew and the general population could explain the increased incidence of skin cancer among cabin crew, and the possible contribution of estimated cosmic radiation.	Exposure to cosmic radiation	Skin cancer	702 cabin crew
Dormanesh B, Vosoughi K, Akhoundi FH, Mehrpour M, Fereshtehnejad SM, Esmaeili S; Sabet AS (2016) [47]	Evaluate the association of being exposed to hyperbaric or hypobaric conditions with carotid artery stenosis and blood flow velocities of cerebral arteries.	Exposure to abnormal pressures	Carotid artery stenosis and blood flow velocities of cerebral arteries	29 divers, 36 pilots and 30 control participants (29 commercial divers (1 female), 31 commercial pilots (5 female), and 30 control participants (5 females))
Grajewski B, Yong LC, Bertke SJ, Bhatti P, Little MP, Ramsey MJ, Tucker JD, Ward EM, Whelan EA, Sigurdson AJ, Waters MA (2015) [28]		Exposure to cosmic radiation	Reproductive hazards and circadian disruption	

**Table 5 ijerph-20-05849-t005:** Risk factors and outcomes of studies reviewed involving both pilots and flight attendants.

Authors (Year)	Objective	Risk Factors	Outcome	Sample
Rafnsson V, Tulinius H, JoÂ nasson JG, Hrafnkelsson J (2001) [48]	Study whether there is increased cancer risk particularly of cancer types previously related to radiation.	Exposure to cosmic radiation	Cancer	1690 cabin attendants, 158 men and 1532 women from the Icelandic Cabin Crew Association and 2 airline companies in Iceland
Bolzán AD, Bianchi MS, Giménez EM, Flaqué MCD íaz, Ciancio VR (2008) [49]	Analyze spontaneous and bleomycin-induced chromosomal aberrations (BLM) in G0 and G2 stages of the cell cycle in peripheral lymphocytes of 21 long-haul aircrew members from Argentina to evaluate BLM-induced clastogenesis as a first approach to determine DNA repair capacity and thereby susceptibility to environmental cancers in aircrew.	Exposure to cosmic radiation	Chromosomal aberrations	21 aircrew members (15 pilots—14 men and one woman—and 6 flight attendants) of international flights from Argentina. Mean age was 48.5
Zeeb H, Hammer GP, Langner I, Schavt T, Bennack S, Blettner M (2009) [50]	Evaluate the incidence of cancer as a function of occupational exposure to radiation.	Exposure to cosmic radiation	Mortality	20,757 cabin crew members
Hammer GP, Blettner M, Langner I, Zeeb H (2012) [51]	Evaluate the incidence of cancer as a function of occupational exposure to radiation.	Exposure to cosmic radiation	Cancer	6000 cockpit crew members
Silva IS, Stavola BD, Pizzi C, Evans AD, Evans SA (2012) [52]	Evaluate the incidence of cancer as a function of occupational exposure to radiation.	Exposure to cosmic radiation	Cancer	16,329 flight crew and 3165 air traffic control officers (ATCOs)
Sultan I, Khayat SK, Garout IR, Alahmadi LS, Alzahrany AAA (2019) [53]	Explore prevalence of otic barotrauma and its risk factors among aircrew members.	Exposure to abnormal air pressures	Otic barotrauma and its risk factors	267 crew members (116 pilots and 151 flight attendants)
Dreger S, Wollschläger D, Hammer TSGP, Blettner M, Zeeb H (2020) [54]	Determine cancer mortality compared with the general population and to examine the dose-response relationship between cumulative occupational radiation dose and specific cancer outcomes in the German aircrew cohort.	Exposure to cosmic radiation	Cancer mortality	26,846 aircrew personnel

**Table 6 ijerph-20-05849-t006:** Quality assessment of studies reviewed involving flight attendants.

Authors (Year)	Study Design	Quality	Score/Completion
Wolf G, Obe G, Bergau L (1999) [21]	Case-control	Newcastle–Ottawa	4 (Moderate evidence)
Haldorsen T, Reitan JB, Tveten U (2001) [22]	Retrospective cohort	Newcastle–Ottawa	4 (Moderate evidence)
Paridou A, Velonakis E, Langner I, Zeeb H, Blettner M, Tzonou A (2003) [23]	Retrospective cohort	Newcastle–Ottawa	4 (Moderate evidence)
Rafnsson V, Sulem P, Tulinius H, Hrafnkelsson J (2003); [24]	Retrospective cohort	Newcastle–Ottawa	4 (Moderate evidence)
Kojo K, Pukkala E, Auvinen A (2005) [25]	Retrospective cohort	Newcastle–Ottawa	6 (Strong evidence)
Pinkerton LE, Waters MA, Hein MJ, Zivkovich Z, Schubauer-Berigan MK, Grajewski B (2012) [26]	Retrospective cohort	Newcastle–Ottawa	3 (Limited evidence)
Pukkala E, Helminen M, Haldorsen T, Hammar N, Kojo K, Linnersj€o A, Rafnsson V, Tulinius H, Tveten Ulf, Auvinen A (2012) [27]	Retrospective cohort	Newcastle–Ottawa	4 (Moderate evidence)
Grajewski B, Whelan EA, Lawson CC, Hein MJ, Waters MA, Anderson JL, MacDonald LA, Mertens CJ, Tseng CY, Cassinelli RT, Luo L (2015) [28]	Retrospective cohort	Newcastle–Ottawa	4 (Moderate evidence)
Johnson CY, Grajewski B, Lawson CC, Whelan EA, Bertke SJ, Tseng CY (2016) [30]	Retrospective cohort	Newcastle–Ottawa	3 (Limited evidence)
Pinkerton LE, Hein MJ, Anderson JL, Little MP, Sigurdson AJ, Schubauer-Berigan MK (2016) [31]	Retrospective cohort	Newcastle–Ottawa	4 (Moderate evidence)
McNeely E, Mordukhovich I, Staffa S, Tideman S, Gale S, Coull B (2018) [32]	Retrospective cohort	Newcastle–Ottawa	4 (Moderate evidence)
Pinkerton LE, Hein MJ, Anderson JL, Christianson A, Dphil MPL, Sigurdson AJ, Schubauer-Berigan MK (2018) [33]	Retrospective cohort	Newcastle–Ottawa	4 (Moderate evidence)

**Table 7 ijerph-20-05849-t007:** Quality assessment of studies reviewed involving pilots.

Authors (Year)	Study Design	Quality	Score/Completion
Band PR, Le ND, Fang R, Deschamps M, Coldman AJ, Gallagher RP, Moody J (1996) [34]	Retrospective cohort	Newcastle–Ottawa	4 (Moderate evidence)
Romano E, Ferrucci L, Nicolai F, Derme V, Stefano GFD (1997) [35]	Retrospective cohort	Newcastle–Ottawa	4 (Moderate evidence)
Haldorsen T, Reitan JB, Tveten U (2000) [36]	Retrospective cohort	Newcastle–Ottawa	4 (Moderate evidence)
Pukkala E, Aspholm R, Auvinen A, Eliasch H, Gundestrup M, Haldorsen T, Hammar N, Hrafnkelsson J, Kyyrönen P, Linnersjö A, Rafnsson V, Storm H, Tveten U (2002) [37]	Retrospective cohort	Newcastle–Ottawa	4 (Moderate evidence)
Zeeb H, Blettner M, Hammer GP, Langner I (2002) [38]	Retrospective cohort	Newcastle–Ottawa	4 (Moderate evidence)
Hajo Zeeb, Maria Blettner, Ingo Langner, Gaël P. Hammer, Terri J. Ballard, Mariano Santaquilani, Maryanne Gundestrup, Hans Storm, Tor Haldorsen, Ulf Tveten, Niklas Hammar, Annette Linnersjö, Emmanouel Velonakis, Anastasia Tzonou1, Anssi Auvinen, Eero Pukkala, Vilhjálmur Rafnsson, Jón Hrafnkelsson (2003) [55]	Retrospective cohort	Newcastle–Ottawa	3 (Limited evidence)
Langner I, Blettner M, Gundestrup M, Storm H, Aspholm R, Auvinen A, Pukkala E, Hammer GP, Zeeb H, Hrafnkelsson J, Rafnsson V, Tulinius H, Angelis GD, Verdecchia A, Haldorsen T, Tveten U, Eliasch H†, Hammar N, Linnersj A (2004) (2003) [39]	Retrospective cohort	Newcastle–Ottawa	5 (Moderate evidence)
Nicholas JS, Butler GC, Davis S, Bryant E, Hoel DG, Mohr Jr. LC, (2003) [40]	Retrospective cohort	Newcastle–Ottawa	3 (Limited evidence)
Pukkala E, Aspholm R, Auvinen A, Eliasch H, Gundestrup M, Haldorsen T, Hammar N, Hrafnkelsson J, Kyyro P, Rafnsson V, Storm H, Tveten U (2003) [41]	Retrospective cohort	Newcastle–Ottawa	4 (Moderate evidence)
Rosenkvist L, Klokker M, Katholm M (2008) [42]	Retrospective cohort	Newcastle–Ottawa	3 (Limited evidence)
Nicholas JS, Swearingen CJ, Kilmer JB (2009) [43]	Retrospective cohort	Newcastle–Ottawa	4 (Moderate evidence)
De Luca JC, Picco SJ, MacIntyre C, Dulout FN, Lopez-Larraza DM (2010) [44]	Retrospective cohort	Newcastle–Ottawa	3 (Limited evidence)
Papailiou M, Mavromichalaki H, Kudela K, Stetiarova J, Dimitrova S (2012) [45]	Retrospective cohort	Newcastle–Ottawa	3 (Limited evidence)
Kojo K, Helminen M, Pukkala E, Auvinen Anssi (2013) [46]	Retrospective cohort	Newcastle–Ottawa	5 (Moderate evidence)
Dormanesh B, Vosoughi K, Akhoundi FH, Mehrpour M, Fereshtehnejad SM, Esmaeili S, Sabet AS (2016) [47]	Cohort study	Newcastle–Ottawa	3 (Limited evidence)
Grajewski B, Yong LC, Bertke SJ, Bhatti P, Little MP, Ramsey MJ, Tucker JD, Ward EM, Whelan EA, Sigurdson AJ, Waters MA (2018) [28]	Case-control	Newcastle–Ottawa	3 (Limited evidence)

**Table 8 ijerph-20-05849-t008:** Quality assessment of studies reviewed involving pilots and flight attendants.

Authors (Year)	Study Design	Quality	Score/Completion
Rafnsson V, Tulinius H, JoÂ nasson JG, Hrafnkelsson J (2001) [48]	Retrospective cohort	Newcastle–Ottawa	4 (Moderate evidence)
Bolzán AD, Bianchi MS, Giménez EM, Flaqué MCD íaz, Ciancio VR (2008) [49]	Retrospective cohort	Newcastle–Ottawa	5 (Moderate evidence)
Zeeb H, Hammer GP, Langner I, SchaVt T, Bennack S, Blettner M (2010) (2009) [50]	Retrospective cohort	Newcastle–Ottawa	3 (Limited evidence)
Hammer GP, Blettner M, Langner I, Zeeb H (2012) [51]	Retrospective cohort	Newcastle–Ottawa	5 (Moderate evidence)
Silva IS, Stavola BD, Pizzi C, Evans AD, Evans SA (2013) (2012) [52]	Retrospective cohort	Newcastle–Ottawa	4 (Moderate evidence)
Sultan I, Khayat SK, Garout IR, Alahmadi LS, Alzahrany AAA (2019) [53]	Cross-sectional	Loney Criterion	5
Dreger S, Wollschläger D, Hammer TSGP, Blettner M, Zeeb H (2020) [54]	Retrospective cohort	Newcastle–Ottawa	3 (Limited evidence)

## Data Availability

Not applicable.

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
