# Peer review of "Aircrew Health: A Systematic Review of Physical Agents as Occupational Risk Factors"

_ijerph, 2023, doi:10.3390/ijerph20105849_

Round 1
Reviewer 1 Report
The authors should explain why they want to focus on physical agent hazards rather than health and safety risks experienced by aircrews in general.
There are very short paragraphs throughout especially in the Introduction and Discussion. In some cases, the segue between paragraphs is questionable.
The authors provide evidence of a number of health effects. However, this is a bit contradictory as they later evaluate the quality of the articles. In other words, if the quality of the article is low or poor, then is it reasonable to report the health effects?
The authors have not made an argument as to why it is important to look specifically at Brazilian aircrew. Does this specific cohort have different exposures and/or risks compared with aircrew from other countries? Similarly, the relevance of the country of origin of the articles included in the review is unclear.
Figure 2 should be in English.
It would have been ideal to provide examples for each main area indicated in Figure 3.
How were key words listed in Figure 4 derived from the main areas of knowledge from Figure 3? Why is noise not found in Table 1 or Figure 4 despite it being mentioned in the Introduction?
How were the quality of studies assessed? Was this done by 1 person, 2 people? If more than 1 individual, how were decisions made if there was a discrepancy? More details are required.
Figure 6 is very difficult to see - suggest enlarging.
It is unclear why the authors grouped the studies in the Results section according to the job category as they did not explain if there is a difference in exposures and/or health outcomes for various air crew. In my opinion, the results should have been presented according to health effect as this is the objective of the review.
Charts 1, 2 and 3 is odd in the sense that the study objectives are listed AFTER the outcome. You cannot have an outcome measure if you do not have an objective.
Section 3.2.3 subheading is for both attendants and pilots yet the first paragraph mentions only pilots. What about results for attendants?
Charts 4, 5 and 6 should indicate the health effects examined in each paper to link back to the objective 1.
Author Response
We appreciate the considerations made. I would like to inform you that the text has been reviewed, taking into account the comments made. Figure 2 has been translated into English and the tables have been adjusted.
Reviewer 2 Report
The review is well done, the authors approach and the methodology are well described. Only the tables are not so easy to read.
It is advisable to re-check the editing
Author Response
We appreciate the considerations made.
Reviewer 3 Report
From my point of view the article is well developed, complying with the format for reporting the results obtained in a systematic review of the literature. I think that the topic addressed is interesting because any study that provides knowledge to control the possible risks that influence the health of workers is of vital importance. I consider that the article can be published with some minor modifications or considerations:
1. Perhaps in the introduction it would be necessary to define what the authors understand by risk factors, physical agents and why they cause health problems.
2. They indicate as justification for the study the lack of literature on the subject. They should be careful, as normally a systematic review is carried out when there is a lot of literature on the subject and an attempt is made to systematise or to know the state of the art of what has been studied.
3. The methods section, from my point of view, should be more structured in order to be better understood, for example, literature search, inclusion and exclusion criteria (this section should include the flow chart of the study selection process) and data extraction.
4. Finally, it should indicate the limitations of the study itself and always, where possible, the limitations of the articles analysed.
From my point of view the article is well developed, complying with the format for reporting the results obtained in a systematic review of the literature. I think that the topic addressed is interesting because any study that provides knowledge to control the possible risks that influence the health of workers is of vital importance. I consider that the article can be published with some minor modifications or considerations:
1. Perhaps in the introduction it would be necessary to define what the authors understand by risk factors, physical agents and why they cause health problems.
2. They indicate as justification for the study the lack of literature on the subject. They should be careful, as normally a systematic review is carried out when there is a lot of literature on the subject and an attempt is made to systematise or to know the state of the art of what has been studied.
3. The methods section, from my point of view, should be more structured in order to be better understood, for example, literature search, inclusion and exclusion criteria (this section should include the flow chart of the study selection process) and data extraction.
4. Finally, it should indicate the limitations of the study itself and always, where possible, the limitations of the articles analysed.
Author Response
We appreciate the considerations made. I would like to inform you that the text has been reviewed, taking into account the comments made.
Round 2
Reviewer 1 Report
The authors have done very little to address the multitude of issues that I brought forward in my initial review.
In the first paragraph, the authors discuss various physical agents such as noise, vibration and temperature. However, the paper does not address any of these physical agents after the introductory paragraph.
Again, there very little rationale why Brazilian pilots and aircrew are of concern. Yes, there are limited studies on this topic from Brazil. But are the results from studies in other countries not representative for Brazilian pilots and aircrew?
In Table 1 where is noise and temperature?
In Figure 4, where is noise and vibration?
Why is section 3.2 important? There is no explanation why the origin of a study warrants a separate section. In my opinion, the authors can simply add the country to Charts 1, 2 and 3 to make the paper an easier read.
In Charts 1, 2 and 3, the authors simply list an outcome; however, there is no context. Were the results statistically significant? What is the relative risk? If cancer is an outcome, is there a lifetime excess cancer risk? In my opinion, this information is critical and would substantially improve the manuscript.
Previously, I suggested that the results should be reported based on the physical agent exposure i.e. pressure, radiation, etc. The authors failed to address why they kept the results according to job cohort. Is there a discernable difference in exposure between pilots and airline crew? If so, this is not clear.
For Charts 4, 5 and 6, for those studies deemed to have 'limited evidence' is there any value in even reporting their results? In other literature reviews that I have read where there is limited evidence, the results are not reported at all because it is difficult to draw conclusions when the strength of evidence is limited. The authors should provide a rationale for inclusion.
In the Discussion, again there is no mention of noise or vibration.
Author Response
Dear reviewer, thank you so much for the notes.
I will try to respond to all the points here. The work prioritized ionizing radiation and abnormal pressures because there is a need for greater clarification for the competent bodies here in Brazil (government agencies and unions). However, the presence of other agents was not completely excluded as the word "Risk Factors" was used in the searches. Articles on noise and vibration were found. However, they were not aligned with the inclusion criteria and, therefore, were not considered.
Section 3.2 is important because it reinforces for national agencies the scarcity of work carried out in Brazil.
The results were separated between pilots and crew precisely to see if there is a significant difference in exposure.